# Automatic Pavement Crack Detection Transformer Based on Convolutional and Sequential Feature Fusion

**DOI:** 10.3390/s23073772

**Published:** 2023-04-06

**Authors:** Zhaoyun Sun, Junzhi Zhai, Lili Pei, Wei Li, Kaiyue Zhao

**Affiliations:** School of Information Engineering, Chang’an University, Xi’an 710064, China

**Keywords:** pavement crack detection, Swin-Transformer, residual network, DETR, low-code, sequence features, convolutional features

## Abstract

To solve the problem of low accuracy of pavement crack detection caused by natural environment interference, this paper designed a lightweight detection framework named PCDETR (Pavement Crack DEtection TRansformer) network, based on the fusion of the convolution features with the sequence features and proposed an efficient pavement crack detection method. Firstly, the scalable Swin-Transformer network and the residual network are used as two parallel channels of the backbone network to extract the long-sequence global features and the underlying visual local features of the pavement cracks, respectively, which are concatenated and fused to enrich the extracted feature information. Then, the encoder and decoder of the transformer detection framework are optimized; the location and category information of the pavement cracks can be obtained directly using the set prediction, which provided a low-code method to reduce the implementation complexity. The research result shows that the highest AP (Average Precision) of this method reaches 45.8% on the COCO dataset, which is significantly higher than that of DETR and its variants model Conditional DETR where the AP values are 36.9% and 42.8%, respectively. On the self-collected pavement crack dataset, the AP of the proposed method reaches 45.6%, which is 3.8% higher than that of Mask R-CNN (Region-based Convolution Neural Network) and 8.8% higher than that of Faster R-CNN. Therefore, this method is an efficient pavement crack detection algorithm.

## 1. Introduction

The maintenance of roads relies greatly on the detection of pavement cracks, which is why automatic detecting techniques are used in practical operation to increase productivity [1]. However, due to the influence of the natural environment, the background noise present in the pavement images is relatively serious. As a result, the accuracy of automatic pavement crack detection algorithms does not meet the requirements of engineering applications. Thus, researchers continue to pay great attention to developing high-accuracy pavement crack detection methods. Cao et al. [2] summarized and compared three types of main automatic pavement crack detection technologies. Among them, the traditional image processing method is affected by the dataset diversity, and its generalization is not good enough. Moreover, in the processing methods of Artificial Intelligence (AI), Machine Learning (ML) still requires manual feature design, while Deep Learning (DL) methods can extract features directly from data through learning and achieve better detection accuracy. In addition, depth information is added to the 3D image processing method, which further improves the detection accuracy. However, it imposes higher requirements on image acquisition equipment and needs a large amount of calculation, which creates a great impact on real-time processing. Therefore, image processing based on DL has become the main method for pavement crack detection.

Pavement crack detection based on DL includes classification-based methods, segmentation-based methods, and object-based methods. For instance, Gopalakrishnan et al. [3] built a classifier based on VGG16 [4] and carried out a transfer learning with the pre-training model on ImageNet [5]. The pavement images were divided into two categories according to the presence of cracks, and the classification accuracy reached 90%. Moreover, Yusof et al. [6] studied different sizes of convolution kernels as well as the number of convolution layers in Deep Convolutional Neural Networks (DCNN). They trained the model on four types of the dataset including transverse crack, longitudinal crack, alligator crack, and non-crack image. As a result, the classification accuracy of the model reached more than 94.25%. In addition, Hammouch et al. [7] used the pre-training model of VGG19 to carry out the transfer learning, making its classification accuracy F1-score of the longitudinal crack, the alligator crack, and the non-crack images more than 90%. However, the classification method could only detect the crack type in the image but was not able to generate either its specific position measure or its size.

Furthermore, Huyan et al. [8] used Unet [9] for semantic segmentation of the pavement cracks, and their method achieved 98.56% and 97.98% for precision and recall, respectively, by optimizing the convolution kernel and the pooling layer. Qu et al. [10] combined classification and segmentation methods by training a classifier using LeNet to filter out the pavement crack images at first and then used VGG16 as the backbone network to extract features that were deconvoluted to get the segmentation result. The final segmentation accuracy F1-Score of the model was improved by 5.2% over Unet on the DeepCrack [11] dataset. Moreover, Song et al. [12] proposed a crackSeg segmentation network, which used Resnet [13] as the backbone for the feature extraction and the dilated convolution for top-layer features. Then, the output results were fused with the bottom features, and the image was reconstructed using the upsampling approach to achieve the segmentation of the pavement cracks. Finally, the model accuracy F1-Score reached 97.92%. In addition, Qu et al. [14] proposed an encoder–decoder segmentation network for pavement cracks where the encoder used the Resnet structure and added an attention module after each residual unit. The multiscale features output from the decoder were deconvoluted and the upsampled results were concatenated. This model achieved better segmentation accuracy than CrackForest [15] when applied to the CFD datasets. The segmentation network used mostly a Unet-like network structure to discriminate whether it belonged to a crack pixel by pixel. Thus, high hardware resources were required when processing large-size images. Moreover, the segmentation results could only generate outlines of the pavement cracks whereas the morphological algorithm would be required to get the location information of the cracks.

The object detection methods, based on DL, could directly obtain the location and category information of the pavement cracks. Jiao et al. [16] summarized the DL object detection algorithms, including the Two-Stage Detection (TSD) method represented by Faster R-CNN [17], and the One-Stage Detection (OSD) method represented by Yolo [18] and SSD [19]. For instance, Ibragimov et al. [20] adopted the Faster R-CNN to process the pavement crack image of 3730 × 10,000 size. The large image was cut into small patches for crack detection at first, and then the detection results were concatenated together. This method could directly generate the location and the type of pavement cracks in the image. Moreover, Shen et al. [21] cascaded the R-CNN network, based on the Faster R-CNN model, to improve the crack detection accuracy. The mAP was improved by 4.1% compared to the base model. In addition, Zhao et al. [22] optimized the Faster R-CNN by combining deformable convolution with a pyramid network as the backbone network, and the mAP of the model was improved by 3.4% compared to it before applying the optimization. Finally, Xiang et al. [23] optimized Yolov5 by adding a transformer [24], which enhanced the model’s ability to extract the feature dependencies of the pavement cracks over a large range. As a result, the detection accuracy F1-Score of the optimized model reached 67.39%.

To sum up, all the above detection methods used CNN as the backbone to extract the features, and then searched the target area by applying the method of the proposal region selection or the anchor mapping, and finally determined the position coordinates and the category information of the pavement cracks by boundary regression and classifier. However, recent studies showed that the Swin-Transformer [25] has more advantages for extracting long-sequence global features. In addition, the detection framework of the DEtection TRansformer (DETR) [26] could directly generate the position coordinates and the categories of objects without any post-processing of the algorithm output, thus reducing the implementation complexity. Liu et al. [27] combined the Swin-Transformer and the DETR and proposed a Multi-Scale Transformer pavement crack detection model named CrackFormerNet. Based on the intermediate output of the Swin-Transformer, the multi-scale features were obtained by the depth-wise convolution, which were fused as input of DETR. The mAP accuracy of the pavement crack detection increased to 84.2%. However, Transformer had a large amount of computation and required a long training time, and the extraction of local features was less efficient than CNN. Hence, in this paper, the Swin-Transformer and the residual CNN were combined to build the backbone to extract the pavement crack features, and a lightweight Transformer detector was designed to propose a more efficient pavement crack detection method. As for the novelty of this paper, it includes the following three aspects:(1)An efficient pavement crack detection method named Pavement Crack DEtection TRansformer (PCDETR) was proposed to improve the detection accuracy by using transfer learning on a small-scale dataset of pavement crack images;(2)The Swin-Transformer and Resnet were combined as two parallel channels of the backbone, which could not only capture the dependency relationship of the long-sequence global features, but also extract the details of the underlying visual local features; the two types of features were fused to enrich the output feature information of the backbone. The network size of the Swin-Transformer could be scaled up and down to meet the needs of different tasks;(3)The lightweight Transformer detection framework was designed to reduce the calculation load. The location and category information of pavement cracks were obtained directly through set prediction, which provided a low-code pavement crack detection method and reduced the implementation complexity.

## 2. Methodology

### 2.1. Overall Structure of PCDETR

An efficient pavement crack detection method, PCDETR, is proposed in this paper. The entire network model consists of a backbone and Lw-Transformer (Lightweight Transformer) detector. The backbone network contains two parallel network channels, the Swin-Transformer and the Resnet, to complete the feature extraction and the fusion of pavement crack images. Moreover, the category and location of these cracks are generated using a Lw-Transformer detector. The overall framework of the model is shown in Figure 1.

Referring to Figure 1, the Swin-Transformer network consists of four stages, where each uses the same basic processing unit structure with a different stacks number, which can scale up and down according to the task difficulty and the hardware capability. In order to improve the model calculation speed, the Swin-Transformer Tiny model is used in Figure 1. The stack number of the Swin-Transformer blocks in each stage are 2, 2, 6, and 2, respectively. The Transformer can obtain the global information of the image, but it requires more resources when processing long-sequence features and it is more computationally intensive, which affects the model inference speed. To address this problem, the backbone network uses the Patch Partition module at first to extract the features of the input image, which uses a convolutional window of 4 × 4 size and outputs a 96-channel local feature map. Then, different Swin-Transformer blocks are used in Stage 1~ Stage 4 to extract the global sequence features of the pavement cracks. The processing of a group of Swin-Transformer Block is as follows: 

Firstly, the regular window attention and the shift window attention are used to extract local features within and between windows;

Then, the Patch Merging module is used to merge the extracted local features to acquire the global feature information of the pavement cracks. Moreover, the Patch Merging procedure reduces the feature map’s length and width by half while doubling the number of channels relative to the input. If the input image size is W × H × 3, the feature map size of the Swin-Transformer output will be W/32 × H/32 × 768 after being processed by four stages as shown in Figure 1. Hence, window attention effectively reduces the amount of computation, and it can support the processing of large-size pavement crack images. 

The backbone has also used Resnet as another channel to get the features extraction of the pavement cracks, as this method can enhance the information transfer by identity mapping, effectively solve the problem of gradient dissipation caused by the increasing network depth and enable a rapid convergence of the deep network model training. Added to that, the convolution operator in Resnet works only on locally adjacent regions of the original image or the input feature graph according to the size of the convolution kernel. Therefore, it has a local feature sensitivity. It can effectively retain the details of local features. To fuse the global sequence features of the Swin-Transformer output with the local features of the Resnet output, it is necessary to ensure a dimensional alignment between the two features. If the input image size is W× H × 3, the feature size of Resnet’s top-level network output will be W/32 ×H/32×512. Consequently, the two features can be concatenated and fused in the channel direction to obtain a W/32 × H/32 × 1280 feature output; this latter will be then processed by the Lw-Transformer detection framework.

In more detail, the Lw-Transformer takes advantage of the Transformer’s good performance in the visual field to provide a concise object detection framework. The simpler the model structure, the faster the calculation speed, but this will reduce the model’s fitting ability to complex problems. During building the model, it was found that when the number of encoders and decoders was less than 3, the convergence rate of the model slowed down significantly. In order to speed up the model processing, this paper reduces the number of encoders and decoders in the Transformer by half, hence using three instead of six. Additionally, the position-encoding information is added to the pavement crack feature map output from the backbone to be the input of the Lw-Transformer detector. The feature map is then converted into a feature sequence using the Transformer’s encoder and decoder to directly output an unordered set of predictions where each element contains the category and the coordinate information of the predicted object. The final Feed-Forward Network (FFN) is a 3-layer perceptron consisting of the ReLU layer, a hidden layer, and the linear mapping layer. The FFN finally predicts the normalized center coordinates as well as the height and width of the pavement crack area, and it uses the softmax function to obtain the pavement crack type.

### 2.2. Double Channel Feature Extractor of the Backbone

#### 2.2.1. Sequential Feature Extractor

The basic processing unit for each stage in the Swin-Transformer consists of two consecutive blocks where the first one adopts a regular Window Multi-head Self-Attention (W-MSA) network structure and the second one adopts a Shifting Window Multi-head Self-Attention (SW-MSA) network structure. The network structure of the Swin-Transformer basic processing unit is shown in Figure 2.

In more detail, a single W-MSA can only extract local features of the fixed Windows but cannot obtain feature dependency between the windows. To solve this problem, SW-MSA uses shifting windows for local feature extraction to enhance information exchange between the different windows. In Figure 2, LN refers to the Layer Norm and MLP represents the two-layer perceptron with an activation function. The above network structure can be expressed as the following equation:(1)$${\hat{z}}^{l}=W-MSA\left(LN\left(z^{l-1}\right)\right)+z^{l-1},\;z^{l}=MLP\left(LN\left({\hat{z}}^{l}\right)\right)+{\hat{z}}^{l},\;{\hat{z}}^{l+1}=SW-MSA\left(LN\left(z^{l}\right)\right)+z^{l},\;z^{l+1}=MLP\left(LN\left({\hat{z}}^{l+1}\right)\right)\;+\;{\hat{z}}^{l+1}$$
where $z^{l}$ is the output of the MLP and ${\hat{z}}^{l}$ is the output of the (S)W-MSA. Moreover, there are residual connections after both MLP and (S)W-MSA, which ensure the information transfer in the deep network.

To extract the feature information between different windows using SW-MSA, the feature map output generated by W-MSA is first shifted to change the window-to-window adjacency. Then, the window attention mechanism is used to extract features on the shifted feature map. If the window size used for the window attention is *M*, the feature map, delivered by the MSA, moves to the right and down according to the steps of M/2, M/2. The window size used by the local attention module in this paper is 7 × 7. Thus, the effect of the window shifting on the feature map is shown in Figure 3.

**Figure 3 sensors-23-03772-f003:**
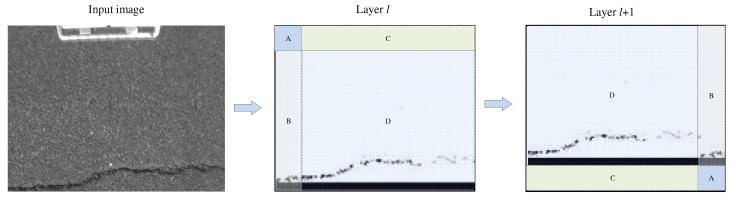
Diagram of window shift on the feature map, where Layer *l* is the feature map output by W-MSA, and Layer (*l* + 1) is the feature map after the window is shifted. In this case, all the positions of regions A, B, C, and D changed. The differences in local features extracted by window attention, before and after the feature map shifting, are shown in Figure 4.

**Figure 4 sensors-23-03772-f004:**
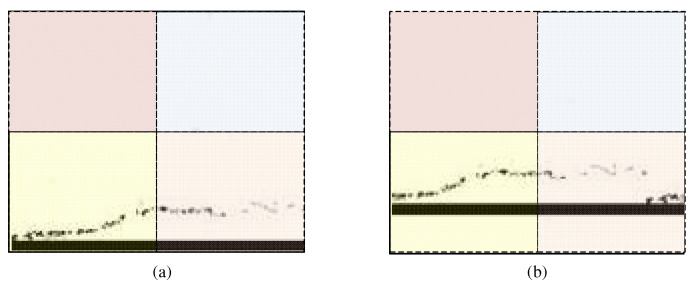
Comparison of feature maps processed by W-MSA and SW-MSA, where (**a**) represents the attention window before the feature map shifted and (**b**) shows the attention window after the feature map shifted. Moreover, the Patch Partition divides each window into four equal patches, and the window attention module extracts the local feature of each area. It can be ensured that information exchange between the different Windows, after repeating the operation of Stage1~Stage4 in the Swin-Transformer, and the global sequence features of the pavement crack can be provided for the subsequent detection network.

#### 2.2.2. Convolutional Feature Extractor

The CNN realizes a fast extraction of local features of the whole image through the local receptive field, the shared weights, and the spatial down-sampling. In terms of local feature representation, it can be an effective supplement to Transformer. However, the introduction of a new network structure will inevitably increase the number of model parameters and the amount of computation. Therefore, in this paper, the balance between the computational cost of the model and the capability of feature representation is considered comprehensively. Hence, Resnet18 is used to construct another channel of the backbone network, and the convolutional features of the pavement crack images are extracted. The residual structure in the backbone network is shown in Table 1.

As can be seen from Figure 5, a residual block has two processing branches for the input data $X_{l}$. The convolutional network structure is shown on the left. The convolutional layers are configured according to the parameters of Conv_2x~Conv_5x in Table 1, and they are connected by Batch Norm and the ReLU layer. On the right is a shortcut which is added to the output of the left branch to form a residual structure. The calculation relation of the residual structure can be expressed as follows:(2)$$Y=h\left(X_{l}\right)+F\left(X_{l}\right)\;\;\;\;\;\;\;\;$$
where $F\left(X_{l}\right)$ represents the convolution network of the left branch in Figure 5, and $h\left(X_{l}\right)$ represents the shortcut of the right branch. When the channel number of $X_{l+1}$ is equal to $X_{l}$, $h\left(X_{l}\right)=X_{l}$. However, at the junction of the different convolution layers (Conv_2x to Conv_5x), the input and output channels’ number of residuals is different. Therefore, a 1 × 1 convolution must be used to improve the dimension of $X_{l}$, to align the feature dimensions of the output of the two branches of the residuals block and to complete the summation operation.

### 2.3. Transformer Detector

In this paper, the encoder–decoder mechanism of the Transformer is used to construct a pavement crack detector, and the feature map generated from the backbone is processed by combining the position-encoding information. The object detection is then converted into a prediction set, and the location and category information of the pavement cracks is directly generated. Compared to CNN networks, the Transformer has a better representation capability for long sequence features. In addition, the Transformer detector no longer uses the Anchor [17] mechanism to search for the proposed region and does not require NMS processing, which simplifies the processing process.

Moreover, to improve model computation efficiency, this paper carries out a lightweight design on Transformer, reducing the six encoders and six decoders in the Transformer to three. Each encoder contains a multi-head self-attention module and a Feed-Forward Network (FFN). To build a deeper model, each module uses a residual connection, followed by a normalized module. In contrast to the encoder, the decoder adds a multi-headed cross-attention module between the multi-headed self-attention module and the FFN module to get the processing results of the encoder. The network structure of the lightweight Transformer is shown in Figure 6.

The first-level encoder converts the feature vector output from the backbone network into a feature sequence as a value, which is added to the spatial position-encoding sequence to generate Query and Key in order to complete the attention calculation of the pavement crack features. Then, the output is processed by FFN to get an encoder processing result. Three encoders are connected one after the other, and the last encoder delivers the processing result to each decoder. As for the decoders, they are also connected step by step, and each decoder calculates the self-attentive weights for the feature output of the higher-level module. Moreover, the results are used together with the output of the encoder to calculate the feature weights of the pavement cracks through the cross-attention. FFN processing is then used to obtain the feature output of the decoder. Finally, based on the last decoder output, the set prediction is performed directly to generate the location and category information of the pavement cracks.

## 3. Experiments and Analysis

The large-size network model can obtain a higher detection accuracy; however, at the same time, it may decrease the inference speed of the system. Given the performance differences among several models, this paper builds PCDETR-T, PCDETR-S, and PCDETR-B backbone based on Swin-T, Swin-S, and Swin-B networks, respectively, and combined with Resnet18 (R18). These proposed models are compared with other ones using the COCO [28] dataset and self-collected pavement crack dataset. The network parameter configuration of Swin-T, Swin-S, and Swin-B is shown in Table 2, describing the number of Swin-Transformer blocks used at each level, the size of the attention Windows, the number of channels for the output features, and the number of heads of multiple self-attention mechanisms.

For example, $\left[\begin{array}{c} win\;size\;7\times 7\\ dim\;96\\ head\;3 \end{array}\right]\times$2 means the attention window size is 7 × 7, the number of channels of the output feature is 96, and the number of heads for the multi-head attention is three. Moreover, the stack number of the Swin-Transformer Blocks is two.

### 3.1. Training Procedure

All training and testing was performed using the same hardware and software environment. The main hardware configurations include: A NVIDIA GPU RTX 2080 Ti, an Intel (R) CPU Xeon (R) Gold 6145 CPU @ 2.00 GHz, and a memory size of 128 GB. As for the software configuration, it includes the use of the Ubuntu 18.04 operating system, the deep learning framework of Python 1.7, the parallel computing framework of cuda10.2, and the GPU driver software version v440.33.

To compare the convergence speed and the detection accuracy rate of the model, the same initial super parameter settings were used in the training process. For example, the learning rate used was 0.0001, the optimizer of the objective function was AdamW, the total number of iterations of the model training was 50 Epoch, the weight attenuation system was 0.0001, the size of the batch size was two, and the probability of the drop-out was 0.1. Finally, random seeds were uniformly set to 42.

### 3.2. Comparison Experiments on the COCO Dataset

To verify the superiority of PCDETR, it is compared to the baseline model DETR and its variant models. All models were trained and tested using the COCO dataset, which is one of the most used benchmarks for object detection. The comparison test results of the detection accuracy of each model are shown in Table 3.

Both the baseline model DETR and its variant model Conditional DETR are presented in Table 3, and they construct the backbone network using Resnet50 (R50) and Resnet101 (R101). In contrast, PCDETR combines the Swin-Transformer and the Resent18 (R18) as the backbone network. It can be seen that the detection accuracy of PCDETR has significantly improved after the replacement of the backbone network. When the parameters are relatively few, the model detection accuracy of PCDETR-T is equivalent to that of the Conditional DETR-R101. This presents a 7.4% increase in AP over the baseline DETR model.

To further study the training speed of PCDETR, Swin-T, Swin-S, and Swin-B were used as backbones for the model training. The loss convergence of the model on the training dataset and the AP value on the verification dataset were analyzed. As the number of training iterations increased, the loss convergence curves of the different models are shown in Figure 7, and the AP change curves are shown in Figure 8.

Referring to Figure 7, PCDETR is relatively easy to train on the large COCO dataset. The convergence rate of the three configuration network models is pretty fast in the initial iteration period; however, in the subsequent training process, the convergence rate tends to be flat. In addition, with the increase of the model size, the loss can also converge lower.

As for Figure 8, the change of the model’s AP curve on the verification set is also consistent with the loss convergence on the training set. The higher the loss of the model is, the lower the AP will be. Moreover, large-size network models can get better detection accuracy.

The COCO training dataset consists of about 118,000 images and the validation dataset consists of 5000 images. It takes about five hours for PCDETR-T to complete an epoch on the training set and about 10 min to complete the processing of all the images regarding the validation dataset. Thus, the processing speed is about 6.3 FPS for the training dataset, and the inference speed is about 8.3 FPS for the validation dataset. As it needs to backpropagate gradients during model training, which is not necessary for model inference, the inference speed is a bit faster. 

### 3.3. Comparison Experiments on the Self-Collection Dataset

The model can obtain more prior knowledge by using training results on large-scale datasets. In this paper, the training results on the COCO data set are used as the pre-training model to initialize PCDETR. PCDETR was applied to pavement crack detection by the transfer learning method. Transfer learning uses a self-collected data set of pavement cracks, including transverse cracks, longitudinal cracks and alligator cracks. The data information on self-collected pavement cracks is shown in Table 4.

The three types of pavement crack images are selected following a ratio of about 1:1:1 to ensure the balance of the number of different types of pavement crack images. Each type of pavement crack image is divided into a training set and a test set, at a ratio of about 4:1. To verify the performance of the PCDETR model for pavement crack detection, it is compared with mainstream object detection algorithms including RetinaNet, Mask R-CNN and Faster R-CNN. The pavement crack detection accuracy and processing speed of different models are shown in Table 5.

As can be seen from Table 5, when the IOU threshold value is 0.5:0.95, the AP accuracy of PCDETR is significantly improved. The AP_75_ value of PCDETR-T is 5.0% higher than that of Mask R-CNN, indicating that PCDETR-T can locate the target more accurately under the condition of a higher IOU requirement. This can effectively avoid problems such as disconnection and incomplete detection of pavement cracks. In terms of processing speed, PCDETR’s processing speed drops due to the large number of parameters in the Transformer. However, the lighter PCDETR-T and Mask R-CNN have roughly the same processing speed.

In addition, the lightest PCDETR-T in Table 5 had the highest AP, surpassing the two larger models. To study the performance difference of PCDETR models using different backbones for pavement crack detection, the three models were compared and analyzed in this paper. The backbone was constructed by combining Swin-B, Swin-S and Swin-T with the respective residual network. The loss curve of each model on the pavement crack training set is shown in Figure 9.

In Figure 9, the convergence of the three models on small-scale pavement crack data is also relatively good, and the overall trend of loss decline is similar to that in Figure 7. Loss decreased rapidly in the initial stage of training, and then tended to be flat. However, the loss curve in Figure 9 is different from that in Figure 7. It is mainly shown that the loss curve of PCDETR-T coincides with that of PCDETR-B, and convergence is lower than that of PCDETR-S with a larger model size. Thus, from the perspective of detection accuracy, the AP value of PCDETR-T is higher. The AP curve of each model is shown in Figure 10. Compared with the AP curve on the COCO dataset, the AP curve on the verification set shows an obvious jitter when the three models are trained on the pavement crack data, revealing that the accuracy stability of the model is not good enough when it is tested for different samples.

After comparative analysis, it was found that the performance difference between the three PCDETR models on the two datasets is mainly caused by the difference in the dataset. Although each model converges well on the two kinds of the training dataset, the advantages of the large model cannot be brought into play on the small-scale pavement crack test dataset, and the accuracy stability of the model is not good enough on the verification dataset.

In Figure 11, the detection results of different models on the same pavement crack images are visualized. The first row is the detection result of the longitudinal crack, the second row is the detection result of the transverse crack, the third row is the detection result of alligator cracks under strong light, the fourth row is the detection result of alligator cracks under normal light, and the fifth row is the detection result of longitudinal cracks with fringe background interference. By comparison, it can be found that the detection results using PCDETR-T are more complete and more accurate. There is no detection fracture, incomplete detection, repeated detection, and other problems.

## 4. Conclusions

In this paper, Swin-Transformer and a residual convolutional neural network are combined to construct the backbone to extract pavement crack features, based on which the lightweight Transformer is designed, and a new pavement crack detection method PCDETR is proposed. The detection accuracy on the COCO dataset exceeds the baseline model, and better detection results are obtained on pavement crack data by transfer learning. Compared with other mainstream object detection methods, the detection accuracy of the proposed method has been significantly improved. In addition, the location and type of pavement crack are obtained directly by set prediction, which reduces the complexity of post-processing. Thus, PCDETR is an efficient method to detect pavement cracks.

However, PCDETR must rely on the prior knowledge of large-scale datasets to obtain better detection accuracy on a small-scale dataset of pavement cracks. Moreover, in the process of model training, the loss has great jitter. Therefore, how to make the model training fast and getting stable convergence on the small-scale pavement crack dataset will become the research target in the next stage.

## Figures and Tables

**Figure 1 sensors-23-03772-f001:**
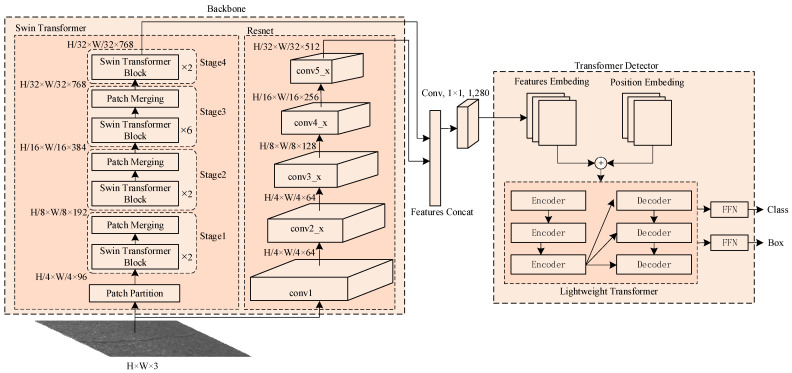
Overall framework of PCDETR.

**Figure 2 sensors-23-03772-f002:**
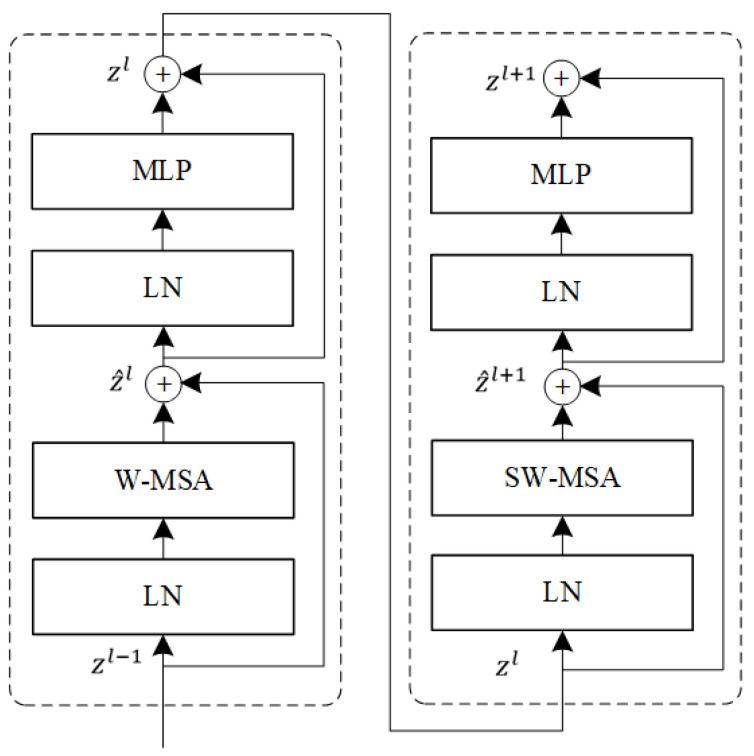
Base processing unit of Swin Transformer Block.

**Figure 5 sensors-23-03772-f005:**
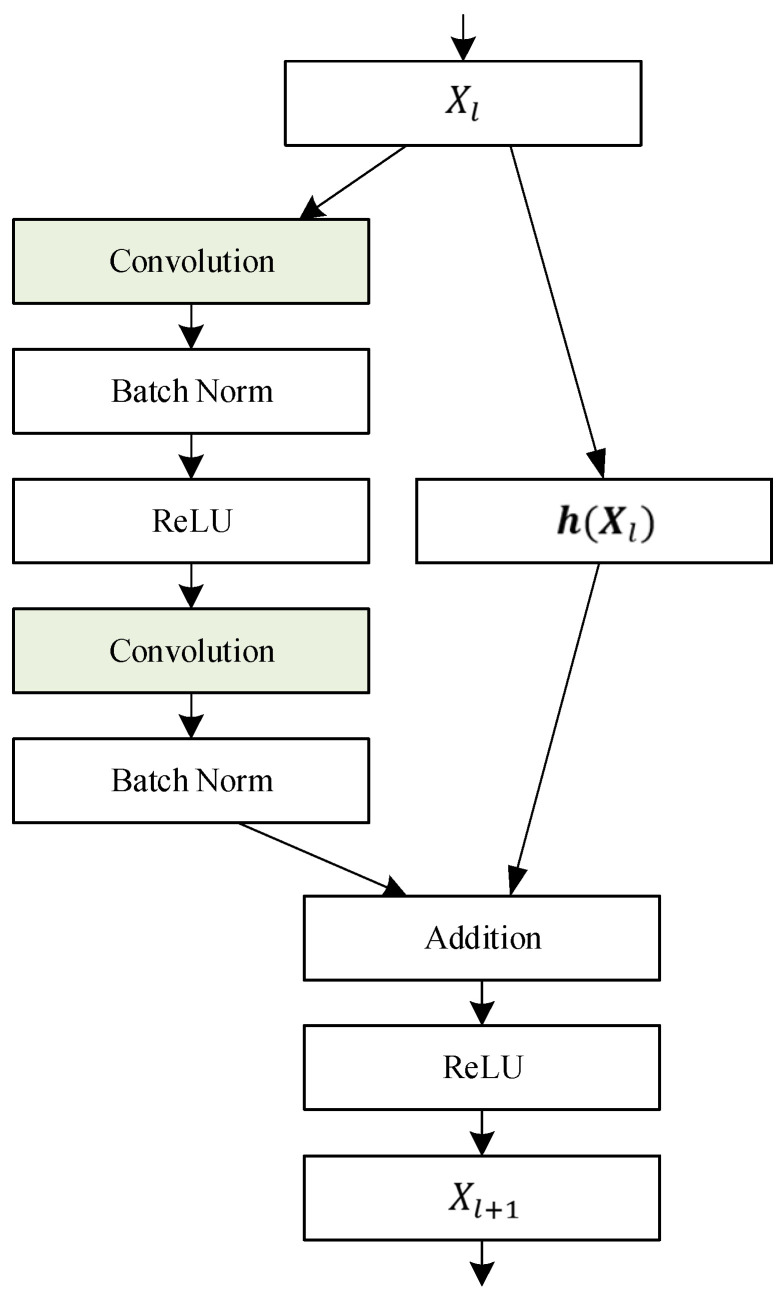
Network structure of a residual block.

**Figure 6 sensors-23-03772-f006:**
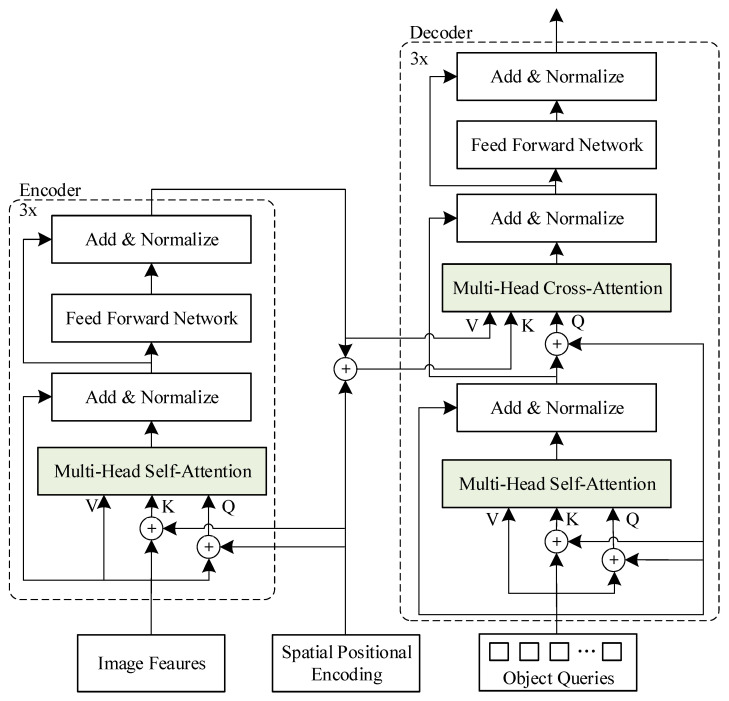
The structure of lightweight Transformer.

**Figure 7 sensors-23-03772-f007:**
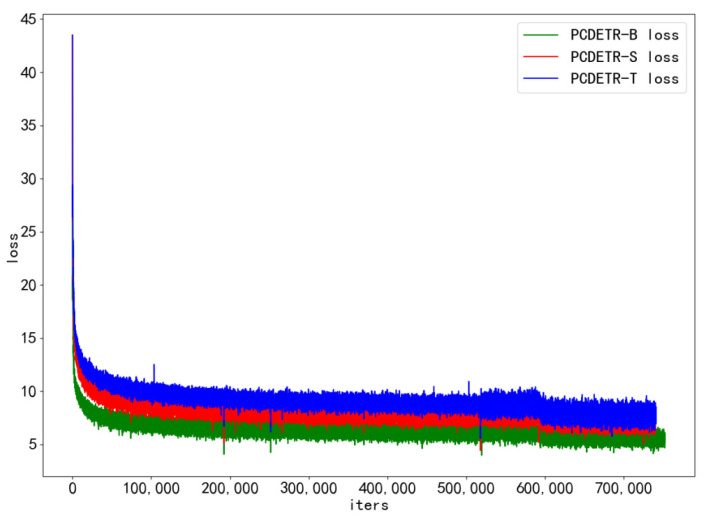
The loss curve of PCDETR with different backbones on the COCO training dataset.

**Figure 8 sensors-23-03772-f008:**
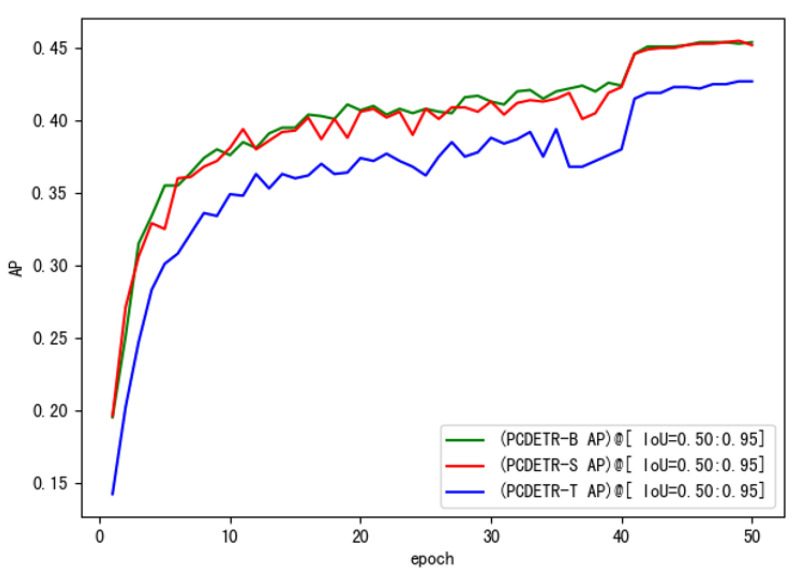
The AP curve of PCDETR with different backbones on the COCO evaluation dataset.

**Figure 9 sensors-23-03772-f009:**
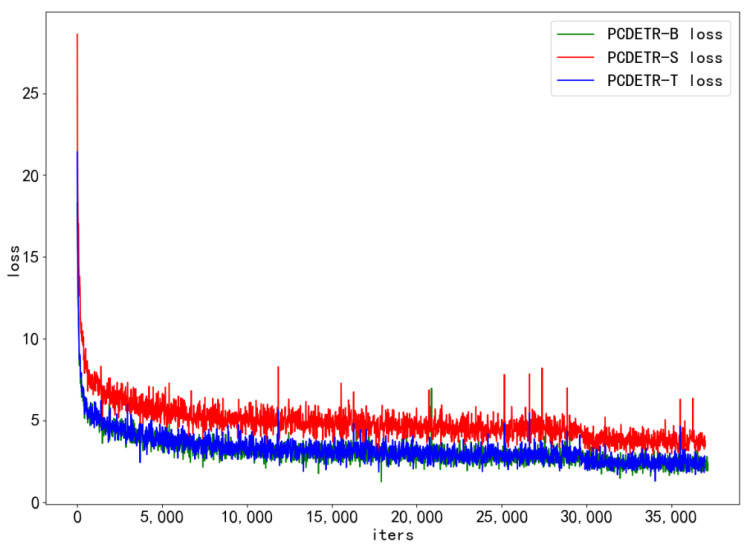
The loss curve of PCDETR with different backbones on the crack training dataset.

**Figure 10 sensors-23-03772-f010:**
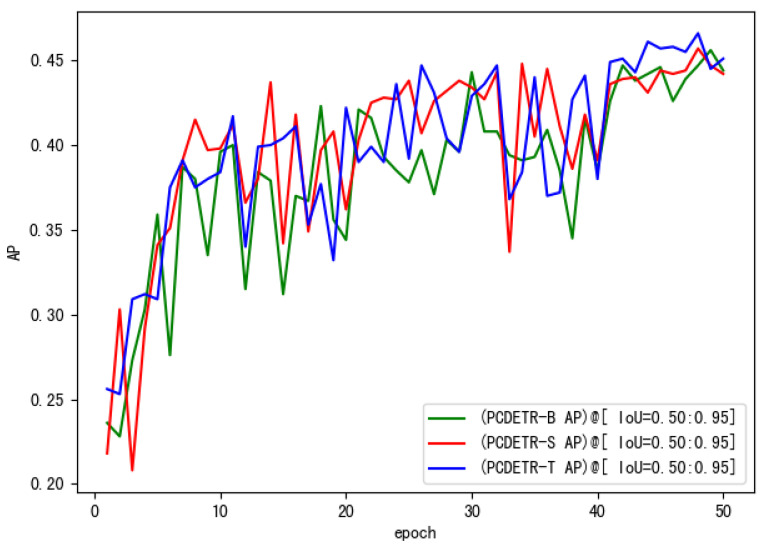
The AP curve of PCDETR with different backbones on the crack evaluation dataset.

**Figure 11 sensors-23-03772-f011:**
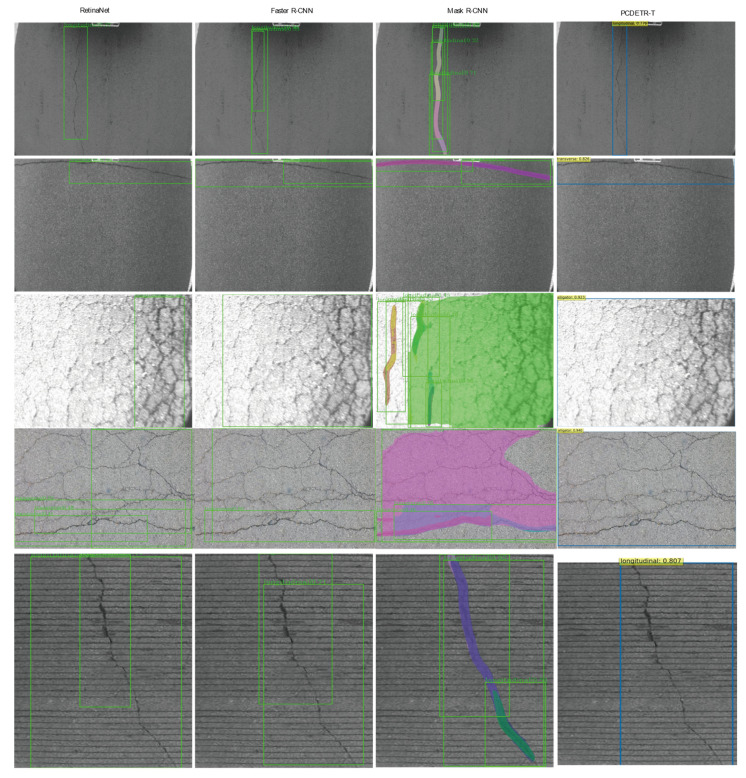
Visualization of pavement crack detection results by each model.

**Table 1 sensors-23-03772-t001:** Resnet Structure of Backbone.

Layer Name	Layer Structure
Conv1	7 × 7, 64, stride 2
Conv2_x	$\left[\begin{array}{c} 3\;\times \;3,\;64\\ 3\;\times \;3,\;64 \end{array}\right]\;\times \;2$
Conv3_x	$\left[\begin{array}{c} 3\;\times \;3,\;128\\ 3\;\times \;3,\;128 \end{array}\right]\;\times \;2$
Conv4_x	$\left[\begin{array}{c} 3\;\times \;3,\;256\\ 3\;\times \;3,\;256 \end{array}\right]\;\times \;2$
Conv5_x	$\left[\begin{array}{c} 3\;\times \;3,\;512\\ 3\;\times \;3,\;512 \end{array}\right]\;\times \;2$

where Conv1 is the first convolution layer, the size of the convolution kernel is 7 × 7, the number of channels is 64, and the step size of the convolution slide window is 2. As for Conv2_x~Conv5_x, they are four residual convolution layers, and each one is composed of two residual blocks. All convolution kernel sizes used by the four residual convolution layers are 3 × 3, and the number of channels of these kernels is 64, 128, 256, and 512, respectively. To sum up, the network structure of a residual block is shown in Figure 5.

**Table 2 sensors-23-03772-t002:** Configuration of Backbone with different Swin transformer block.

Stage	Swin-T	Swin-S	Swin-B
1	$\;\left[\begin{array}{c} win\;size\;7\times 7\\ dim\;96\\ head\;3 \end{array}\right]\times$2	$\left[\begin{array}{c} win\;size\;7\times 7\\ dim\;96\\ head\;3 \end{array}\right]\times$2	$\left[\begin{array}{c} win\;size\;7\times 7\\ dim\;96\\ head\;4 \end{array}\right]\times$2
2	$\left[\begin{array}{c} win\;size\;7\times 7\\ dim\;192\\ head\;6 \end{array}\right]\times$2	$\left[\begin{array}{c} win\;size\;7\times 7\\ dim\;192\\ head\;6 \end{array}\right]\times$2	$\left[\begin{array}{c} win\;size\;7\times 7\\ dim\;192\\ head\;8 \end{array}\right]\times$2
3	$\left[\begin{array}{c} win\;size\;7\times 7\\ dim\;384\\ head\;12 \end{array}\right]\times 6$	$\left[\begin{array}{c} win\;size\;7\times 7\\ dim\;384\\ head\;12 \end{array}\right]\times 18$	$\left[\begin{array}{c} win\;size\;7\times 7\\ dim\;384\\ head\;16 \end{array}\right]\times 18$
4	$\left[\begin{array}{c} win\;size\;7\times 7\\ dim\;768\\ head\;24 \end{array}\right]\times 2$	$\left[\begin{array}{c} win\;size\;7\times 7\\ dim\;768\\ head\;24 \end{array}\right]\times 2$	$\left[\begin{array}{c} win\;size\;7\times 7\\ dim\;768\\ head\;32 \end{array}\right]\times 2$

**Table 3 sensors-23-03772-t003:** The detection-precision comparison results of each model on COCO.

Model	Params (M)	AP (%)	AP_50_ (%)	AP_75_ (%)	AP_S_ (%)	AP_M_ (%)	AP_L_ (%)
DETR-R50	41	34.9	55.5	36.0	14.4	37.2	54.5
DETR-R101	60	36.9	57.8	38.6	15.5	40.6	55.6
Conditional DETR-R50	44	40.9	61.8	43.3	20.8	44.6	59.2
Conditional DETR-R101	63	42.8	63.7	46.0	21.7	46.6	60.9
**PCDETR-T**	**48**	**42.3**	**63.1**	**44.9**	**21.8**	**46.9**	**62.6**
**PCDETR-S**	**70**	**44.7**	**65.8**	**47.9**	**23.3**	**51.3**	**64.9**
**PCDETR-B**	**108**	**45.8**	**66.5**	**48.5**	**24.7**	**51.5**	**65.2**

where AP represents the average detection precision when the Intersection Over Union (IOU) threshold ranges between 0.5 and 0.9 (with a step size of 0.05). Moreover, the AP_50_ represents the average detection precision when the IOU threshold is 0.5, AP_75_ represents the average detection precision when the threshold of IOU is 0.75, AP_S_ represents the average detection precision of small targets, AP_M_ represents average detection precision of medium targets, and AP_L_ represents average detection precision of large targets.

**Table 4 sensors-23-03772-t004:** The self-collected pavement crack dataset.

Dataset	Transverse	Longitudinal	Alligator	Total	Ratio
Train	1673	1680	1820	5173	0.817
Test	415	355	392	1162	0.183
Total	2088	2035	2212	6335	1

**Table 5 sensors-23-03772-t005:** The detection precision comparison results of each model on the pavement crack dataset.

Model	Backbone	AP(%)	AP_50_ (%)	AP_75_ (%)	FPS
Faster R-CNN	Resnet50	36.8	61.2	38.0	17.3
RetinaNet	Resnet50+FPN	22.0	41.8	20.4	18.0
Mask R-CNN	Resnet50	41.8	72.4	42.6	14.8
**PCDETR-T**	**Swin-Tiny + R18**	**45.6**	**68.7**	**47.6**	**15.9**
**PCDETR-S**	**Swin-Small + R18**	**44.8**	**66.8**	**47.3**	**11.3**
**PCDETR-B**	**Swin-Base + R18**	**44.9**	**67.9**	**47.8**	**8.8**

## Data Availability

Some or all data, models, or code that support the findings of this study are available from the corresponding author upon reasonable request.

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
