# Peer review of "Automatic Pavement Crack Detection Transformer Based on Convolutional and Sequential Feature Fusion"

_sensors, 2023, doi:10.3390/s23073772_

Round 1
Reviewer 1 Report
It is a good study and works on improving automatic crack detection which is much needed in the realm of pavement rehabilitation investigations. The paper is structured well and presents results clearly. The results are also of value to the research community. The use of English can be improved, as an example the authors use the word 'therefore' 18 times.
Author Response
Thank you very much for taking your time in reviewing our manuscript and conducting careful checking. This problem has been revised. The whole article has been polished. For more detailed information,please see the attachment.
For clarity, all the modified and extended parts of the text are marked in blue in the revised manuscript.

Reviewer 2 Report
The authors present a comprehensive study of automatic pavement crack detection.
The novelty of the paper is clearly pointed by 3 aspects: a new method to improve the detection accuracy by using transfer learning on smallscale dataset of pavement crack images is proposed; b) two parallel channels of the backbone were combined and fused to enrich the output feature information of the backbone; the lightweight Transformer detection framework was designed to reduce the calculation load.
The manuscript reports a substantial body of results and is a well-done job. This is an exciting and well-written article and I recommend it to be published if the editor agrees.
Author Response
Thank you very much for taking your time in reviewing our manuscript and conducting careful checking.
The whole article has been polished again. For more detailed information, please see the attachment.
For clarity, all the modified and extended parts of the text are marked in blue in the revised manuscript.

Reviewer 3 Report
1.When carrying out the lightweight design on Transformer,why reducing six encoders and decoders in the Transformer to three.
2. The image performance related to network structure is not good. The structure of lightweight transformer looks messy because of too many mechanics.
3. How to balance the computational efficiency and precision by ensuring that the computational speed of the two parallel structures is close to each other?
4.There is a lack of comparison between the training and verification time of the proposed framework.
5. Some words in the text have problems with italics and font size , which need to be corrected.
Round 2
Reviewer 3 Report
I have no comments on this version. it can be acceptted.